# Proteome Analysis of Bevacizumab Intervention in Experimental Central Retinal Vein Occlusion

**DOI:** 10.3390/jpm13111580

**Published:** 2023-11-07

**Authors:** Lasse Jørgensen Cehofski, Anders Kruse, Mads Odgaard Mæng, Benedict Kjaergaard, Jakob Grauslund, Bent Honoré, Henrik Vorum

**Affiliations:** 1Department of Ophthalmology, Odense University Hospital, 5000 Odense, Denmark; jakob.grauslund@rsyd.dk; 2Biomedical Research Laboratory, Aalborg University Hospital, 9000 Aalborg, Denmark; benedict.kjaergaard@rn.dk; 3Department of Clinical Research, University of Southern Denmark, 5000 Odense, Denmark; 4Department of Ophthalmology, Aalborg University Hospital, 9000 Aalborg, Denmark; anders.kruse@rn.dk (A.K.); m.maeng@rn.dk (M.O.M.); 5Department of Biomedicine, Aarhus University, 8000 Aarhus, Denmark; bh@biomed.au.dk; 6Department of Clinical Medicine, Aalborg University, 9000 Aalborg, Denmark

**Keywords:** retina, retinal vein occlusion, proteome, proteomics, mass spectrometry, vascular endothelial growth factor, biomarker, bevacizumab

## Abstract

Bevacizumab is a frequently used inhibitor of vascular endothelial growth factor (VEGF) in the management of macular edema in central retinal vein occlusion (CRVO). Studying retinal protein changes in bevacizumab intervention may provide insights into mechanisms of action. In nine Danish Landrace pigs, experimental CRVO was induced in both eyes with argon laser. The right eyes received an intravitreal injection of 0.05 mL bevacizumab (n = 9), while the left control eyes received 0.05 mL saline water (NaCl). Retinal samples were collected 15 days after induced CRVO. Label-free quantification nano-liquid chromatography–tandem mass spectrometry identified 59 proteins that were regulated following bevacizumab treatment. Following bevacizumab intervention, altered levels of bevacizumab components, including the Ig gamma-1 chain C region and the Ig kappa chain C region, were observed. Changes in other significantly regulated proteins ranged between 0.58–1.73, including for the NADH-ubiquinone oxidoreductase chain (fold change = 1.73), protein-transport protein Sec24B (fold change = 1.71), glycerol kinase (fold change = 1.61), guanine-nucleotide-binding protein G(T) subunit-gamma-T1 (fold change = 0.67), and prefoldin subunit 6 (fold change = 0.58). A high retinal concentration of bevacizumab was achieved within 15 days. Changes in the additional proteins were limited, suggesting a narrow mechanism of action.

## 1. Introduction

Retinal vein occlusion is the second most common retinal vascular disease (RVO). Only diabetic retinopathy is more frequent. In Europe, the number of individuals affected by RVO is estimated to be 900,000 and is expected to reach 1.1. million by 2050 [1]. RVO is subdivided into central retinal vein occlusion (CRVO) and branch retinal vein occlusion (BRVO). In CRVO, the entire retina is affected by the occlusion, while a sectorial area in affected in BRVO. Central retinal vein occlusion (CRVO) is a common retinal vascular disease caused by the impaired outflow of the central retinal vein, the major outflow vessel of the eye, while BRVO results from the impaired outflow of a retinal branch vein [2,3]. Objective findings in RVO include flame-shaped hemorrhages, macular edema, and optic-disc swelling. Optic-disc swelling is particularly prominent in CRVO [3,4,5]. 

Visual loss in RVO mainly results from complications of the condition, including macular edema, macular ischemia, vitreous hemorrhage due to neovascularizations, and neovascular glaucoma [3,5,6,7]. Another significant challenge in the management of RVO is the development of inner retinal atrophy following ischemia. Although macular edema resolves upon anti-VEGF therapy, a subgroup of patients exhibit atrophy in the neuroretina. The atrophy has been reported to be pronounced in the retinal inner plexiform layer and the outer nuclear layer [8].

BRVO is more frequent than CRVO. In general, CRVO is considered the most severe subtype of RVO, as the entire retina is affected. Macular edema secondary to CRVO is the most common cause of visual loss in both BRVO and CRVO [5,6,7,8,9], and visual acuity rarely improves above 20/40 if the macular edema is left untreated [10]. Formation of macular edema secondary to CRVO occurs through a complex multifactorial mechanism. Occlusion of the central retinal vein results in increased resistance to blood flow in retinal arterioles, leading to retinal hypoxia. Retinal hypoxia triggers an increased production of vascular endothelial growth factor A (VEGF-A) and a complex inflammatory response promoted through interleukin (IL)-6, IL-8, IL-18, S100A12, fibrinogen, fibronectin, galectin-3, and monocyte chemotactic protein-1 [11,12,13,14,15]. Elevated retinal levels of VEGF-A and pro-inflammatory proteins increase the vascular permeability driving the accumulation of fluid in the macula [12].

Macular edema is effectively treated with intravitreal anti-VEGF injections as first-line therapy, while dexamethasone intravitreal implants are considered a second-line treatment [16,17,18,19]. Although there have been significant advances in the management of CRVO, macular edema secondary to CRVO remains a significant burden to patients. Approximately 50% of CRVO patients need anti-VEGF injections four years after CRVO is diagnosed [20], and the growing number of intravitreal injections results in significant pressure on retina clinics [21]. Hunt and co-workers [22] recently reported the 3-year outcomes in a cohort of 527 eyes with CRVO. Half of all the eyes continued to receive anti-VEGF therapy at 36 months. Only 62 out of the 527 eyes (12%) achieved resolution of the macular edema without treatment for more than 6 months.

Bevacizumab is a full-length monoclonal immunoglobulin G (IgG) antibody that binds and inhibits all isoforms of VEGF-A [23]. Studying protein changes downstream of VEGF-A neutralization by bevacizumab may elucidate novel mechanisms that mediate the beneficial response to anti-VEGF treatment. The objective of a proteome study is to identify and quantify the entire set of proteins in a given tissue or body fluid [24]. Here, we report on changes to the retinal proteome following bevacizumab intervention in experimental CRVO. 

## 2. Materials and Methods

### 2.1. Animal Preparation

The experiments were approved by the Danish Animal Experiments Inspectorate, permission no. 2019-15-0201-01651. The approval included intravitreal bevacizumab intervention in experimental CRVO. A total of 9 Danish Landrace pigs, weighing 30–40 kg, were used for the experiments and housed under a 12 h light/dark cycle. The animals were anesthetized with an intramuscular injection of Zoletil (Virbac, Carros, France) (ketamine 6.25 mg/mL, tiletamine 6.25 mg/mL, zolazepam 6.25 mg/mL, xylain 6.25 mg/mL, and butorphanol 1.25 mg/mL). During anesthesia the animals were observed by a veterinarian or an anesthesiologist for vital parameters and if needed the trachea was intubated and the animal was manually ventilated until the animal was awake again. Topical anesthesia was performed with Oxybuprocaine Hydro 0.4% (Bausch & Lomb, Kingston Upon Thames, UK) and Tetracaine (Bausch & Lomb, Kingston Upon Thames, UK). Dilation of the pupils was performed with Tropicamide 1.0% (Bausch & Lomb, Kingston Upon Thamas, UK) and Phenylephrine 10% (Bausch & Lomb, Kingston Upon Thames, UK) [25]. After the last operation, the animals were eventually euthanized with an overdose of pentobarbital (Le Vet, Oudewater, The Netherlands).

### 2.2. Experimental Central Retinal Vein Occlusion (CRVO)

Experimental CRVO was induced in both eyes of the animals as previously described [26]. In brief, CRVO was induced in close proximity to the optic nerve head with a standard argon laser (532 nm) given by indirect ophthalmoscopy using a 20D lens. Laser settings were 400 mW, with an exposure time of 550 ms. A total of 30–40 laser applications were used per occlusion. By applying the laser directly on retinal veins at the optic nerve head, thrombotic material was displaced toward the optic nerve head and the lamina cribrosa, causing an experimental CRVO. Experimental CRVO was considered successful when stagnation of venous blood and the development of flame-shaped hemorrhages appeared.

An intravitreal injection of 0.05 mL bevacizumab 25 mg/mL (Avastin; Roche, Copenhagen, Denmark) was injected into the right eyes of the animals (n = 9), while an intravitreal injection of 0.05 mL sodium chloride 9 mg/mL (NaCl) (B. Braun, Denmark) was given to the left control eyes (n = 9). Successfully induced CRVO was confirmed by angiography 4 days after the induction of CRVO.

Fifteen days after experimental CRVO, the eyes were enucleated and dissected on ice under a microscope. The animals were euthanized immediately after enucleation. The anterior segment of the eye was removed, the vitreous body was removed by aspiration into a 5 mL syringe, and the neurosensory retina was separated from the RPE/choroid complex with tweezers and stored at −80 °C until further use.

### 2.3. Sample Preparation for Mass Spectrometry

Label-free quantification nano-liquid chromatography–tandem mass spectrometry (LFQ nLC-MS/MS) was performed to compare CRVO + bevacizumab vs. CRVO + NaCl. Sample preparation was performed as follows: The samples were mixed with lysis buffer consisting of 5% sodium dodecyl sulfate and 50 mM triethyl ammonium bicarbonate (TEAB). The samples were sonicated on ice with a QSonica Sonicator Q125 (QSonica, Newtown, CT, USA) with previously described settings [27].

A Direct Detect Spectrometer (Merck KGaA, Darmstadt, Germany) was used to measure the protein concentration of each sample by infrared spectrometry [27]). Up to 100 µg of protein from each sample used for proteomic analysis was prepared by the suspension-trapping sample preparation method [28] using micro S-Trap spin columns (Protifi, Farmingdale, NY, USA). Samples were alkylated with Tris(2-carboxyethyl)phosphine hydrochloride (TCEP) at 95 °C for 10 min, and then iodoacetamide was added for a further 30 min at room temperature, protected from light. The solution was centrifuged and the supernatant was transferred to a new tube. Phosphoric acid was added and mixed to a final concentration of 1.2%. Then, S-Trap binding buffer (90% aqueous methanol, 100 mM TEAB) was added to the sample, which was then added to the S-Trap column and centrifuged. Then, the sample was washed three times with S-Trap buffer. An amount of 20 µL tryptic digestion buffer was added to the column, which was capped to limit evaporative loss and finally incubated at 37 °C overnight. The next day, the peptides were eluted in three centrifugation steps by first adding 40 µL of 50 mM TEAB, then 40 µL of 0.2% formic acid and finally 35 µL of 50% acetonitrile with 0.2% formic acid. Each of the elutions were pooled, and dried in a vacuum centrifuge. They were then resuspended in 100 mM TEAB followed by measurement of the peptide concentration by fluorescence, as described in a previous paper [27]. Finally, the samples were dried and dissolved at 1 µg of peptide per µL in 0.1% formic acid, and 1 µg was injected.

### 2.4. Label-Free Quantification by Nano Liquid Chromatography–Tandem Mass Spectrometry (LFQ nLC-MS/MS)

LFQ nLC-MS/MS was performed on an Orbitrap Fusion Tribrid mass spectrometer equipped with an EasySpray ion source coupled to a Dionex Ultimate^TM^ 3000 RSLC nanosystem (Thermo Fisher Scientific Instruments, Waltham, MA, USA). Peptide separation was performed as previously described [26], with 500 mm analytical columns (500 mm × 75 µm PepMap RSCL, C18, 2 µm, 100 Å, Thermo Scientific) applying a 122 min gradient by mixing buffer A (99.9% water, 0.1% formic acid) with buffer B (99.9% acetonitrile, 0.1% formic acid). Mass spectrometry was performed with the Universal Method for label-free quantification. One µg of each sample was injected. The Orbitrap was used to obtain full scans at a range of 375–1500 *m*/*z* at a resolution of 120,000. The automatic gain control target was set to 4 × 10^5^, and the maximum injection time was set to 50 ms. The quadrupole was used to isolate precursor ions with a window of 1.6 *m/z*. Collision-induced dissociation was performed with an energy of 35%. The ion trap was used to detect MS^2^ scans, with an automatic gain control target of 2 × 10^3^ and a maximum injection time of 300 ms. MaxQuant software version 1.5.7.4 [29] (Max Planck Institute of Biochemistry, Martinsried, Germany; https://maxquant.net/maxquant/) was used to search the raw data files against the Uniprot *Sus scrofa* and *Homo sapiens* databases. 

### 2.5. Filtration of Proteins and Statistics

MaxQuant output files (Appendix A) were uploaded to the Perseus software, version 1.6.2.3 [30] (Max Planck Institute of Biochemistry, Martinsried, Germany; https://maxquant.net/perseus/). Quantitative values were log2-transformed, and technical replicates were averaged, and at least two unique peptides were required for successful identification. Furthermore, successful identification and quantification were required in at least 70% of the samples in each group. The Student’s *t*-test was performed in Perseus to compare CRVO + bevacizumab vs. CRVO + NaCl. Proteins were considered significantly regulated if *p* < 0.05.

## 3. Results

### 3.1. Filtration of Proteins and Statistics

CRVO was successfully induced in all eyes (Figure 1). Angiography confirmed successfully induced CRVO (Figure 2). 

### 3.2. Retinal Proteome Changes Following Bevacizumab Intervention in Experimental CRVO

A total of 2423 proteins were successfully identified in the combined set of samples (Appendix A). In total, 2027 retinal proteins were successfully identified in at least 70% of the samples in each group (Appendix A), and statistical analysis was performed on these proteins. A total of 59 proteins were significantly regulated in bevacizumab intervention in CRVO (Figure 3) (Table 1). A total of 25 proteins were upregulated following bevacizumab intervention, while 34 proteins were downregulated (Figure 3) (Table 1). 

High concentrations of components of bevacizumab were observed in retinas treated with bevacizumab, including the Ig gamma-1 chain C region (fold change = 48.4; *p* = 5.83 × 10^−9^) and the Ig kappa chain C region (fold change = 29.1; *p* = 3.68 × 10^−10^). Among the upregulated proteins, the fold changes ranged between 1.08–1.73 (Table 1). Among the downregulated proteins, the fold changes ranged between 0.58–0.93 (Table 1).

Proteins with the most pronounced upregulations included the NADH-ubiquinone oxidoreductase chain (fold change = 1.73; *p* = 0.011), protein transport protein Sec24B (fold change = 1.71; *p* = 0.027), glycerol kinase (fold change = 1.61; *p* = 0.0020), and agrin (fold change = 1.41; *p* = 0.031) (Table 1). Proteins with the most pronounced downregulation included prefoldin subunit 6 (fold change = 0.58; *p* = 0.044), guanine nucleotide-binding protein G(T) subunit gamma-T1 (fold change = 0.67; *p* = 0.024), lipopolysaccharide-responsive and beige-like anchor protein (fold change = 0.68; *p* = 0.0050), and F-actin-capping protein subunit alpha-1 (fold change = 0.68; *p* = 0.037 (Table 1).

## 4. Discussion

Our study reports on proteome changes related to bevacizumab intervention in an experimental model of CRVO. A model of non-ischemic CRVO was chosen for the study as the majority of CRVO cases are known to be non-ischemic [3]. Alternatively, an ischemic CRVO can be created by occluding four branch retinal veins in the porcine eye, but stages with severe ischemia may be more resistant to anti-VEGF therapy and may not be well-suited for studies of anti-VEGF intervention [31]. The CRVO model is discussed in detail in a recent report [31], but a number of advantages and drawbacks are relevant to highlight. The porcine retina is well suited for studies of retinal vascular diseases [31,32]. The porcine retina is fully vascularized, like the human retina, and is similar to the human retina in size and photoreceptor distribution. The porcine retina has a cone photoreceptor-dense area, the visual streak, which is localized superior to the optic nerve head, but it does not have a macular area like the human retina. 

Our study used a dose of 1.25 mg intravitreal bevacizumab, which is the standard dose for the treatment of macular edema secondary to CRVO. The dose of 1.25 mg intravitreal bevacizumab is also the dose used in the SCORE2 randomized trial, which compared the efficacy of bevacizumab with the anti-VEGF agent aflibercept [33]. Different doses of bevacizumab could be tested in the model to assess dose-dependent retinal proteome changes. Furthermore, bevacizumab could be tested head-to-head against other anti-VEGF agents or against dexamethasone intravitreal implants. However, the use of porcine eyes is associated with a number of limitations, including the high level of expenses per animal, including high housing costs [31,34]. Therefore, it would be highly expensive to conduct the experiments with large sample sizes or to test different doses of bevacizumab since a head-to-head comparison would double the number of animals needed. Another limitation in the use of porcine eyes is the growth of the animals. Providing a series of interventions like, for example, loading with three doses at intervals of 4 weeks is difficult to perform in pigs, as the young animals undergo substantial growth within few months [31].

The study has a number of clinical implications. High concentrations of bevacizumab components were reached 15 days after intervention, consistent with successful drug delivery to the retina. In addition to the bevacizumab components, changes in other proteins were modest, with fold changes ranging between 0.58–1.73. It is questionable if the small changes in these proteins have biologically relevant functions. A broad variety of retinal proteome changes downstream of VEGF signaling could potentially have been expected. However, the proteome analysis indicated that bevacizumab does not result in a broad retinal response, and the main effect of bevacizumab may be limited to the retinal vasculature. On the contrary, the proteome analysis indicated a narrow mechanism of action, which is important to keep in mind in the clinical setting. In the case of insufficient clinical response to bevacizumab, clinicians should consider switching therapeutic strategies, considering the multifactorial nature of macular edema in CRVO [15]. Our findings in this study are similar to our previous report on aflibercept intervention in experimental CRVO, where a narrow mechanism of action of aflibercept was identified [26]. Our previous studies of dexamethasone intravitreal implants have demonstrated a wider mechanism of action associated with dexamethasone intervention [17,19]. The broader mechanism of action following dexamethasone intravitreal implants must be considered in the case of a limited response to bevacizumab therapy.

Pro-inflammatory as well as anti-inflammatory features have previously been described [35,36,37,38]. However, no pro-inflammatory or anti-inflammatory features associated with bevacizumab were observed in our study, and no intraocular reaction was observed in the animals.

It is a limitation that the entire neuroretina was studied. The neuroretina has high complexity due to its multiple layers and cell types. Future studies of anti-VEGF therapy in experimental CRVO may be beneficial for conducting studies that are more specifically directed at the retinal vasculature; for example, by isolating retinal veins affected by the occlusion.

The narrow mechanism of action of bevacizumab highlighted in our study should be considered in the context of the multifactorial pathogenesis of CRVO. A major issue in the management of CRVO is that the treatment predominantly addresses complications of the condition [4,12]. However, CRVO is a complex condition associated with several risk factors that may also influence the course of the condition [39]. 

Age, hypertension, diabetes, glaucoma, and an increased cup-disc ratio are important risk factors for CRVO [4]. Females are less likely to be identified with CRVO compared with males [39], and the prevalence of RVO in general rises sharply with age [40]. Hypertension is a well-established risk factor for CRVO, and the risk increases further with end-organ damage due to hypertension. While diabetes is a well-known risk factor for CRVO, the complexity of diabetes as a risk factor is higher in CRVO. Diabetes as a risk factor for CRVO is particular important in the case of metabolic syndrome with hypertension, diabetes, and hyperlipidemia combined. Furthermore, the risk of CRVO increases in the case of diabetes with end-organ disease [39]. Peripheral artery disease and prior stroke are also associated with an increased risk of CRVO [39]. Open-angle glaucoma and a large cup-disc ratio are also associated with the development of CRVO [39,41]. Uveitis may also be complicated by RVO, especially in the case of infectious uveitis [42]. In young adults, mental disorders were recently reported as a risk factor for RVO in a Korean national cohort. More specifically, depression, anxiety disorder, and sleep disorder were associated with an increased risk of RVO. These mental disorders remained significantly associated with an increased risk of RVO after correction for smoking and metabolic syndrome [43], but the mechanisms behind the association remain poorly understood.

## 5. Conclusions

High levels of bevacizumab components were observed in the CRVO model, consistent with high retinal concentrations of bevacizumab being reached within a period of 15 days. In addition to changes in bevacizumab components, retinal proteome changes following bevacizumab therapy were very modest and unlikely to have biological implications. Overall, the proteome analysis suggests that bevacizumab has a narrow mechanism of action. This narrow mechanism of action is essential to consider in cases of insufficient response to bevacizumab therapy.

## Figures and Tables

**Figure 1 jpm-13-01580-f001:**
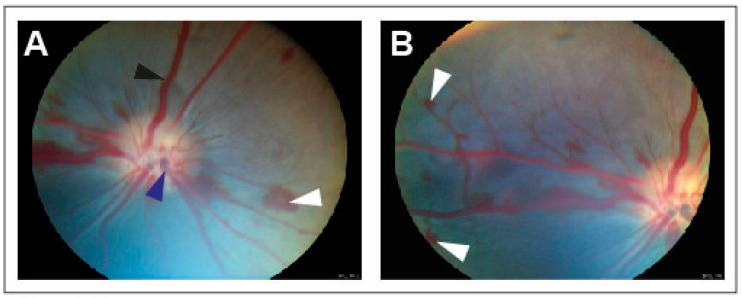
Fundus images obtained within 30 min of induced CRVO. (**A**) Fundus image of experimental CRVO with venous dilation and retinal hemorrhages upstream of the occlusion. White arrow: Retinal hemorrhage. Black arrow: Dilated retinal vein. Dark blue arrow: Site where laser is applied to displace thrombotic material toward the lamina cribrosa. (**B**) Peripheral view of retinal hemorrhages. White arrow: Retinal hemorrhage.

**Figure 2 jpm-13-01580-f002:**
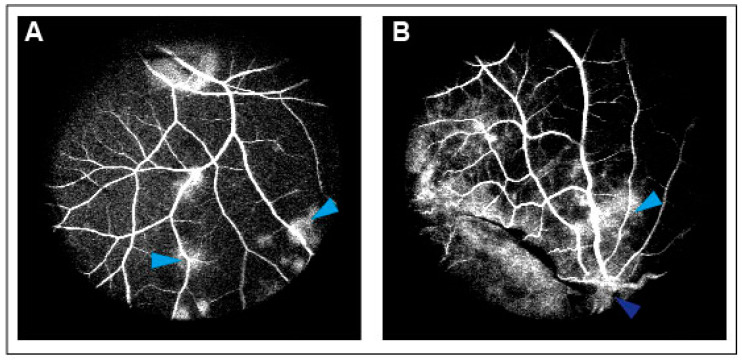
(**A**) Leakage of fluorescein observed following CRVO. Light blue arrow: Leakage of fluorescein. (**B**) Leakage of fluorescein upstream of the occlusion site. Light blue arrow: Leakage of fluorescein. Dark blue arrow: Site of occlusion.

**Figure 3 jpm-13-01580-f003:**
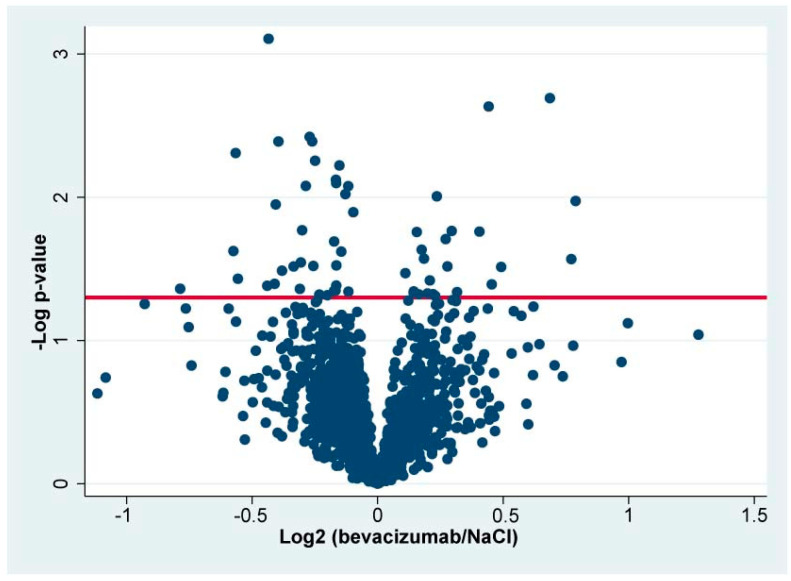
Volcano plot. The log2 LFQ ratio (bevacizumab/NaCl) is plotted on the *x*-axis. On the *y*-axis, the -log *p*-value refers to the log10-transformed *p*-value from the *t*-test used to test if a protein was significantly regulated. The horizontal line denotes a significance level of 0.05. Statistically significantly regulated proteins are located above the horizontal line. Components of bevacizumab are not included in the volcano plot.

**Table 1 jpm-13-01580-t001:** Significantly regulated proteins ordered according to fold change.

Protein ID	Protein Names	Gene Names	*p*-Value	Fold Change
P0DOX5	Ig gamma-1 chain C region	IGHG1	6 × 10^−9^	48.4
P01834	Ig kappa chain C region	IGKC	4 × 10^−10^	29.9
O79874	NADH-ubiquinone oxidoreductase chain	MT-ND1	0.011	1.73
O95487-2	Protein transport protein Sec24B	SEC24B	0.027	1.71
P32189-1	Glycerol kinase	GK	0.002	1.61
O00468-6	Agrin	AGRN	0.031	1.41
Q9UPA5	Protein bassoon	BSN	0.041	1.37
P10155	60 kDa SS-A/Ro ribonucleoprotein	TROVE2	0.0020	1.36
Q16518	Retinoid isomerohydrolase	RPE65	0.017	1.32
Q9UGP8	Translocation protein SEC63 homolog	SEC63	0.046	1.24
Q15437	Protein -transport protein Sec23B	SEC23B	0.017	1.23
Q96D71-3	RalBP1-associated Eps domain-containing protein 1	REPS1	0.030	1.21
Q8IX01-4	SURP and G-patch domain-containing protein 2	SUGP2	0.020	1.21
P36873	Serine/threonine-protein phosphatase PP1-gamma catalytic subunit	PPP1CC	0.010	1.18
Q8TB36-2	Ganglioside-induced differentiation-associated protein 1	GDAP1	0.049	1.17
Q99719	Septin-5	SEPT5	0.048	1.17
O43837	Isocitrate dehydrogenase [NAD] subunit beta, mitochondrial	IDH3B	0.038	1.15
O94906-2	Pre-mRNA-processing factor 6	PRPF6	0.047	1.15
Q12979-4	Active breakpoint cluster region-related protein	ABR	0.027	1.14
Q6L8Q7-2	2,5-phosphodiesterase 12	PDE12	0.023	1.13
A6NHR9-2	Structural maintenance of chromosomes flexible hinge domain-containing protein 1	SMCHD1	0.048	1.12
P00571	Adenylate kinase isoenzyme 1	AK1	0.017	1.11
Q9P265	Disco-interacting protein 2 homolog B	DIP2B	0.045	1.10
O95155-2	Ubiquitin-conjugation factor E4 B	UBE4B	0.046	1.10
Q00610-2	Clathrin heavy chain 1	CLTC	0.034	1.08
P46776	60S ribosomal protein L27a	RPL27A	0.013	0.93
O95865	N(G),N(G)-dimethylarginine dimethylaminohydrolase 2	DDAH2	0.046	0.92
P62258	14-3-3 protein epsilon	YWHAE	0.0080	0.92
O43242	26S proteasome non-ATPase regulatory subunit 3	PSMD3	0.010	0.91
P26234-2	Vinculin	VCL	0.024	0.90
Q8WXA9-2	Splicing regulatory glutamine/lysine-rich protein 1	SREK1	0.006	0.90
Q9UEY8-2	Gamma-adducin	ADD3	0.030	0.89
Q9NPF4	Probable tRNA N6-adenosine threonylcarbamoyltransferase	OSGEP	0.041	0.89
P35221	Catenin alpha-1	CTNNA1	0.0080	0.89
Q92784-4	Zinc finger protein DPF3	DPF3	0.0080	0.89
P08195-2	4F2 cell-surface antigen heavy chain	SLC3A2	0.045	0.89
P05198	Eukaryotic translation initiation factor 2 subunit 1	EIF2S1	0.020	0.89
Q99536	Synaptic vesicle membrane protein VAT-1 homolog	VAT1	0.045	0.89
Q96T51-2	RUN and FYVE domain-containing protein 1	RUFY1	0.048	0.87
Q19S50	Signal transducer and activator of transcription 3	STAT3	0.048	0.85
P62495-2	Eukaryotic peptide chain release factor subunit 1	ETF1	0.0060	0.84
Q64L94	Proteasome-activator complex subunit 1	PSME1	0.030	0.84
O75190-2	DnaJ homolog subfamily B member 6	DNAJB6	0.0040	0.83
Q9NQ48	Leucine zipper transcription factor-like protein 1	LZTFL1	0.0040	0.83
Q863I2	Serine/threonine-protein kinase OSR1	OXSR1	0.0080	0.82
Q9BY07-6	Electrogenic sodium bicarbonate cotransporter 4	SLC4A5	0.017	0.81
Q06210-2	Glutamine-fructose-6-phosphate aminotransferase 1	GFPT1	0.029	0.81
Q9NVH1-3	DnaJ homolog subfamily C member 11	DNAJC11	0.044	0.81
A3KMH1-2	von Willebrand factor A domain-containing protein 8	VWA8	0.030	0.79
P14136-3	Glial fibrillary acidic protein	GFAP	0.033	0.77
Q13394	Protein mab-21-like 1;Protein mab-21-like 2	MAB21L1	0.0040	0.76
Q9NYF8-2	Bcl-2-associated transcription factor 1	BCLAF1	0.011	0.75
O75369-2	Filamin-B	FLNB	0.040	0.75
P12931	Proto-oncogene tyrosine-protein kinase Src	SRC	8 × 10^−4^	0.74
P49959-2	Double-strand break repair protein MRE11A	MRE11A	0.041	0.74
P52907	F-actin-capping protein subunit alpha-1	CAPZA1	0.037	0.68
P50851-2	Lipopolysaccharide-responsive and beige-like anchor protein	LRBA	0.0050	0.68
P63211	Guanine nucleotide-binding protein G(T) subunit gamma-T1	GNGT1	0.024	0.67
O15212	Prefoldin subunit 6	PFDN6	0.044	0.58

## Data Availability

Please refer to Appendix A.

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
