# Peer review of "Proteome Analysis of Bevacizumab Intervention in Experimental Central Retinal Vein Occlusion"

_jpm, 2023, doi:10.3390/jpm13111580_

Round 1
Reviewer 1 Report
Comments and Suggestions for Authors
The paper Proteome Analysis of Bevacizumab Intervention in Experimental Central Retinal Vein Occlusion addresses the consequences of retinal vein occlusion and protein level change for better pathological pathways understanding and treatment of Central Retinal Vein Occlusion (CRVO). The authors are very skilled in what they do and report. Authors have a significant record of publications in the discussed topic. The results of this study are very needed for clinicians and drug developers for CRVO. Experimental design is very simple, clear and accurately suites to achieve the aim authors set for this experiment.
The results section is very well structured and illustrated. It gives clear understating of upregulated and down regulated proteins after vein occlusion itself and Bevacizumab treatment as a rescue drug.
However, some improvements can be made to this manuscript:
1. Some schematic representation of the disease pathogenesis pathways in the discussion section would be favorable based on the obtain experimental results. This also includes discussion of the pathogenesis of Central Retinal Vein Occlusion (CRVO). Depending on the pathogenesis CRVO can be managed differently. The exact cause of CRVO can vary, and in many cases it is multifactorial. The etiology of CRVO is complex and can be influenced by various risk factors. Some of the main reasons and risk factors associated with CRVO include:
· Aging and smoking: One of the primary risk factors for CRVO is age. The condition is more common in individuals over the age of 50. Smoking is a known risk factor for various vascular diseases, including those affecting the eye. It can contribute to the development of CRVO.
· Vascular Disease and Thrombophilia: Underlying vascular diseases, such as atherosclerosis and hypertension, can increase the risk of CRVO by damaging blood vessels and promoting clot formation. Certain blood clotting disorders or genetic factors that make the blood more prone to clotting can increase the risk of CRVO.
· Diabetes: People with diabetes are at a higher risk of developing vascular problems, including CRVO. Diabetic retinopathy can lead to changes in the blood vessels of the retina.
· Glaucoma: Glaucoma, a condition characterized by increased intraocular pressure, can affect blood flow in the retina and may contribute to CRVO.
· Hyperlipidemia: Elevated levels of cholesterol and other lipids in the blood can lead to the buildup of plaque in blood vessels, potentially increasing the risk of CRVO.
· Inflammatory Conditions: Inflammatory conditions like vasculitis can affect the blood vessels in the eye and increase the risk of CRVO.
· Medications: Some medications, such as oral contraceptives and certain hormones, may increase the risk of blood clot formation and, consequently, CRVO in susceptible individuals.
· Systemic Diseases: Conditions like multiple sclerosis and systemic lupus erythematosus (SLE) have been associated with an increased risk of CRVO.
· Hyperviscosity Syndromes: conditions that increase the viscosity of the blood can increase the risk of CRVO.
Several intersecting molecular pathways, genetic and environmental bottlenecks in the CRVO formation can lead to impactful insights with the results obtained by the authors in the discussion section.
Reference 17: it is arguable to make a reference for the unpublished material.
12 out of 30 references are for Cehofski first author papers. Probably it is reasonable in this case, because author is referring to previously published methods, but it is a little too much self-citation in my opinion.
Reviewer 2 Report
Comments and Suggestions for Authors
Summary:
The authors of this paper have conducted research to elucidate the mechanism of action of Bevacizumab, a commonly employed inhibitor of vascular endothelial growth factor (VEGF), in the treatment of macular edema associated with central retinal vein occlusion (CRVO). The topic is particularly relevant and engaging due to the challenges posed by this medical condition.
The paper is structured in a highly organized fashion, skillfully utilizing tables and figures to facilitate the comprehension of the presented data.
Observations:
- In the Methods section, the authors repeatedly reference their previous publications for the methods used. While it's understandable that they might assume readers are familiar with their work, providing a brief description of the methods used in this study would enhance the paper's comprehensibility. It's important for readers to have a clear understanding of the methodology, especially if they are not intimately familiar with the authors' previous work.
- The authors did not attempt to test alternative doses or treatment periods, which could have added valuable insights to the study. Providing a rationale for why these specific parameters were chosen and discussing the potential implications of alternative dosing or treatment regimens would strengthen the paper.
- The bibliography primarily consists of the authors' own publications. While this is not necessarily a problem if those publications are highly relevant, it's crucial to recommend a more comprehensive review of the literature. This would demonstrate a broader understanding of the field and strengthen the paper's theoretical foundation.
- The Discussion section could benefit from greater depth. Expanding on the implications of the study's findings, discussing potential limitations, and suggesting avenues for future research would provide readers with a more comprehensive understanding of the paper's significance.
- Clinical Relevance: The paper is described as lacking clinically relevant results. Expanding on this point by specifying what clinical implications were expected but not observed in the study, and discussing the potential reasons behind this, would provide clarity and insight into your assessment.
- Line 134: "aflibercept" The manuscript does not mention aflibercept elsewhere. Please revise and correct the misstype.
- Line 137: What statistical test was used?

Reviewer 3 Report
Comments and Suggestions for Authors
This manuscript has scientific merit but the authors need to perform the following modifications before it can be accepted for publication:
Ethical approval: Apparently, the ethical approval was obtained in 2019, for a different study. The authors should include that they have obtained permission to include this study under the same ethical approval from 2019.
The authors need to try gene ontology and pathway analyses to try to get an idea of the pathways involved.
Since this is a personalized medicine journal, the authors need to bring their study into the context of the journal. In the introduction, introduce the importance of bevacizumab as a personalized medicine. In the discussion, discuss the application of the results in personalized medicine.
The authors need to discuss the inflammatory role of bevacizumab. Please check and cite these references in breast cancer, Bevacizumab was shown to induce inflammation 10.1016/j.cellsig.2018.11.007 https://pubmed.ncbi.nlm.nih.gov/30445167/ . The same has been reported for intraocular inflammation following intravitreal bevacizumab injection doi: 10.1136/bjo.2009.166033 and doi: https://bjo.bmj.com/content/93/4/457 https://bjo.bmj.com/content/94/4/525.1 and https://bjo.bmj.com/content/93/4/457 . Other studies reported that Bevacizumab alleviates inflammation doi: 10.3389/fphar.2018.00649 https://www.frontiersin.org/articles/10.3389/fphar.2018.00649/full . The authors should discuss their results in light of any gene ontology and pathways results they will obtain and the inflammatory roles of Bevacizumab if any.
Comments on the Quality of English LanguageMinor editing of English language required
Round 2
Reviewer 2 Report
Comments and Suggestions for Authors
The manuscript was extensively revised by authors and was improved considerably, according to the recommendations.
Reviewer 3 Report
Comments and Suggestions for Authors
The authors have answered my comments.
I suggest that this manuscript be accepted.
Comments on the Quality of English LanguageEnglish language is appropriate. Proofread for minor typos.